# Mollaret’s Meningitis due to Herpes Simplex Virus 2: A Case Report and Review of the Literature

**DOI:** 10.3390/microorganisms12071363

**Published:** 2024-07-03

**Authors:** Liliana Gabrielli, Isabella Banchini, Evangelia Petrisli, Giulia Piccirilli, Simona Venturoli, Matteo Pavoni, Alessia Cantiani, Federica Lanna, Caterina Campoli, Matteo Montironi, Maddalena Giannella, Tiziana Lazzarotto

**Affiliations:** 1Microbiology Unit, IRCCS Azienda Ospedaliero-Universitaria di Bologna, 40138 Bologna, Italy; liliana.gabrielli@aosp.bo.it (L.G.);; 2Department of Medical and Surgical Sciences, Section of Microbiology, University of Bologna, 40138 Bologna, Italy; 3Infectious Diseases Unit, IRCCS Azienda Ospedaliero-Universitaria di Bologna, 40138 Bologna, Italy; 4Department of Medical and Surgical Sciences, Infectious Diseases Section, Azienda Ospedaliero-Universitaria di Bologna, 40138 Bologna, Italy

**Keywords:** Mollaret’s meningitis, herpes simplex virus 2, aseptic meningitis, acyclovir

## Abstract

Mollaret’s meningitis is a rare neurological disorder characterized by recurrent episodes of aseptic lymphocytic meningitis, often associated with herpes simplex virus 2 (HSV-2) infection. We report the case of a 39 y.o. Italian woman who experienced four episodes of aseptic lymphocytic meningitis between 2004 and 2023, diagnosed as Mollaret’s meningitis. In each episode, the patient presented with fever, severe headache and photophobia. In two episodes cutaneous vesicles in the left gluteal area preceding meningitis symptoms were also reported. A diagnostic evaluation included a physical–chemical analysis and a real-time PCR of the cerebrospinal fluid (CSF). The CSF presented pleocytosis with lymphocytic predominance and a positive HSV-2 load, with a peak of 1234 copies/mL. The patient was treated successfully with acyclovir, and the symptoms resolved without neurological sequelae. This case highlights the importance of comprehensive diagnostic testing and vigilant monitoring to manage Mollaret’s syndrome effectively.

## 1. Introduction

Mollaret’s meningitis (MM) is a neurological syndrome characterized by recurrent aseptic lymphocytic meningitis episodes that usually last for 2–7 days with a favorable outcome but unpredictable recurrences, often associated with herpes simplex virus 2 (HSV-2) infection.

It was first mentioned by Pierre Mollaret in 1944, who described recurrent aseptic meningitis in three patients with similar neurological symptoms and absence of a bacterial etiology [1]. The French neurologist observed the presence of “fantomes cellulaires” (cell ghosts), a type of large endothelial-like cell, in the cerebrospinal fluid (CSF). The increasing use of molecular diagnostic techniques has highlighted HSV-2 as the most commonly isolated pathogen in this condition. Other viruses that can rarely lead to the syndrome are HSV-1 [2,3,4], varicella zoster virus [5], Epstein–Barr virus [6], human herpesvirus-6 [7] and enterovirus [8].

MM is also called benign recurrent aseptic meningitis, benign recurrent endothelial meningitis and benign recurrent endothelial–leukocytic meningitis. Using electron microscopy, de Chadarévian and Becker [9] demonstrated how Mollaret’s endothelial-like cells were monocytic in origin. Typically, patients present with headache, meningism, and photosensitivity. Once the meningitis resolves, the patient has no neurologic sequelae until the next episode. Symptom-free intervals last from a few weeks to many years, with some patients experiencing only three episodes and others reporting over 30 episodes [10].

The definition of MM varies among studies; in a recent cohort study, a minimum of only two episodes of meningitis were applied to diagnose MM instead of three as usually required. The authors, only considering HSV-2-associated MM, reported an annual incidence of 1.2 cases per 1,000,000 adults [10].

This present paper describes clinical features and laboratory findings in a patient who presented with four episodes of meningitis over a twenty-year period; a literature review of the last decade is also included.

## 2. Case Presentation

In October 2023, a 39 y.o. Italian woman was admitted to the Emergency Department of IRCCS Azienda Ospedaliero-Universitaria of Bologna, reporting, from the day before, intense headache, neck stiffness and photophobia. Her temperature was 37.2 °C, her blood pressure was 116/75 mmHg, her heart rate measured 84/min, and she had a pulse oxygen saturation of 98% on room air. A physical examination revealed an erythematous cutaneous rash over the neck, in the thoracic and abdominal region, without signs of herpetic lesions, although the patient reported the presence of cutaneous vesicles in the left gluteal area 10 days before.

The patient’s medical history included hypothyroidism, latex and penicillin allergies, allergic asthma and a gastro-esophageal reflux disease. In addition, the patient reported recurrent meningitis episodes that occurred in 2004, 2013 and 2016. An urgent lumbar puncture was performed, revealing limpid CSF with normal pressure. The CSF profile showed pleocytosis with a predominance of lymphocytes and a normal glucose concentration (Table 1). A syndromic multiplex PCR on CSF was performed (FilmArray ME Panel, BioFire Diagnostics, Salt Lake City, UT, USA), confirming the clinical suspicion of HSV-2 meningitis in the absence of bacterial etiology. Antiviral therapy was promptly started (acyclovir 750 mg i.v. every 8 h) in association with an antihistaminic for the neck rash, and the patient was transferred to the Infectious Disease Unit. The quantitative PCR (HSV-2 ELITe MGB^®^ kit, ELITech Group, Torino, Italy) on CSF revealed a viral load of 1234 copies/µL, whereas HSV-2 PCR on blood was negative. Imaging findings were normal in the absence of endocrine hypertension and ischemic or hemorrhagic areas. A progressive reduction in photophobia and neck rigidity, in conjunction with the resolution of the headache, were described in the following 7 days of hospitalization. Subsequently, the patient reported mild hyposthenia in the left side of the body, drowsiness and dizziness. The symptoms disappeared after antihistaminic therapy suspension. Laboratory parameters revealed mild leukocytosis (10.6 K/µL; range 3.6–10.5) with a normal differential, hemoglobin (15 g/dL) and platelet count (263 K/µL).

During medical work-up, the patient reported that similar symptoms had occurred during the previous three hospitalizations, specifically mild fever (37.2–37.5 °C), headache, neck stiffness and photophobia.

In the first episode, which occurred in 2004, the clinical examination showed the presence of Brudzinski and Kernig signs in the absence of cutaneous vesicles. Although no molecular testing was performed in CSF, Pandy’s test was positive, indicating hyperproteinorrachia. The whole blood count showed increased white blood cells (9.29 K/µL; range 4.8–8.5). Serum antibodies for the coxsackie virus, echovirus, poliovirus and adenovirus were negative. Serological tests for EBV, cytomegalovirus and *Borrelia burgdorferi* revealed IgG-positive and IgM-negative results. The electroencephalography was normal. The patient was discharged with a diagnosis of lymphocytic meningitis.

The patient returned to the emergency department in 2013 and in 2016 presenting intense headache, back pain and photophobia persisting from the day before. At admission, the patient’s temperature was 37.2 °C. No cutaneous vesicles were reported in the episode of 2013, while in 2016, they were present 7 days before the onset of the symptoms in the left gluteal area. On both occasions, urgent lumbar puncture was performed, and a positive result for HSV-2 was detected, with a low viral load (<500 copies/mL), as reported in Table 1. Blood tests showed leukocytosis in both episodes (11.71 K/µL; range 4.8–8.5 and 12.94 K/µL; range 3.6–10.5, respectively) with a normal differential count.

Several diagnostic tests were performed during the four hospitalizations, showing positive HSV-2 IgG antibodies and negative hepatitis C virus and human immunodeficiency virus antibodies. Imaging findings were normal through all the episodes in the absence of endocrine hypertension, hypodensity areas and ischemic or hemorrhagic areas.

During the first episode, in 2004, Ceftriaxone and corticosteroids were administered, and the patient was discharged after 9 days without any symptoms. Antiviral therapy was administered (acyclovir i.v. 750 mg every 8 h) during the other episodes. In 2013 and 2016, the HSV-2 CSF positive PCR result allowed for an interruption of antibiotic therapy, which is empirically administered by protocol when signs of meningitis are present. No antibiotic therapy was administered in the last episode.

CSF analysis during the entire study period is outlined in Table 1, including pleocytosis, elevated protein levels and normal or decreased glucose levels consistent with aseptic meningitis.

## 3. Discussion

In 1962, Bruyn et al. published the criteria for the diagnosis of MM as recurring episodes presenting with severe headache, meningismus and fever in the absence of a detectable etiological agent [11]. After the introduction of molecular methods, many studies reported HSV-2 as a frequent etiological agent of MM and analyzed characteristic clinical features in depth. Thereafter, Bruyn’s criteria were revised and modified. In 2020, Gadhiya et al. [12] defined its main characteristic features, in particular (1) recurrent episodes of aseptic meningitis; (2) absence of symptoms between episodes; (3) spontaneous remission of symptoms; (4) transient neurological symptoms in 50% of patients; (5) absence of neurological sequelae; (6) HSV-2 as the main etiological agent; and (7) genital herpes in 50% of cases.

The case herein described met the diagnostic criteria specified above. In particular, the patient experienced three episodes of HSV-2 meningitis and one episode of presumed, but not microbiologically confirmed, viral meningitis in a period of 20 years. The CSF profile was typical for aseptic meningitis, with mildly elevated protein levels, low glucose and leukocyte count >5 cells/mm^3^ with a lymphocyte prevalence. To our knowledge, this is the first time that the CSF viral load in the different MM episodes was reported. The HSV-2 load was very low, ranging from less than the lower limit of quantification of RT PCR (500 copies/mL) and 1234 copies/mL.

The occurrence of HSV-2 MM was described by several authors, and in Table 2, 16 case reports of immunocompetent patients are reported that have been published in the last ten years [12,13,14,15,16,17,18,19,20,21,22,23,24,25,26,27]. In these patients, the CSF parameters presented a similar pattern characterized by pleocytosis and, in most cases, hypoglycorrhachia and hyperproteinorrachia as well. The number of episodes described (Table 2) ranged from 2 to 12. In this regard, Petersen et al. reported that the risk of recurrence is higher for patients that experienced more than three previous episodes. A higher prevalence in female (70%) than in male patients and a mean age of 40 years was also reported [10].

An association between HSV-1 and MM has been described in rare cases; the first report was by Steel et al. [2], who identified HSV-1 in the CSF of a patient with four episodes of aseptic meningitis. However, viral meningitis in immunocompetent adults is usually caused by HSV-2, while HSV-1 is more commonly associated with encephalitis [28].

The basis for the susceptibility to develop MM in some individuals is still unknown. Franzen-Röhl et al. observed that patients with recurrent HSV-2 meningitis have increased in vitro, HSV-specific, adaptive and innate immune responses compared to healthy HSV-2 seropositive blood donors, raising the possibility of immune-mediated pathology in the development of MM [29]. The role of genetic factors in developing MM was also investigated. A genetic mutation in the TLR3, UNC-93B gene has been observed in some patients. This mutation is involved in regulating inflammation and may cause an overactive immune response and increased inflammation in the brain, causing recurrent meningitis episodes [13]. Recently, it was suggested that single-gene inborn errors of immunity could contribute to the inability to maintain HSV-2 latency in sensory ganglia, thereby predisposing certain people to recurrent meningitis. In particular, the gene variants involved were related to the autophagy pathway [21]. In addition, autoimmune disorders like systemic lupus erythematosus have been linked to MM [30].

In our patient, cutaneous vesicles in the left gluteal area occurred 10 days before the onset of meningitis in two episodes. This is in agreement with the observation that a history of genital herpes occurs in less than 50% of patients [12]. Overall, latency in dorsal root ganglia is presumed to be the source of recurrencies leading to meningitis [16].However, HSV-2 is currently being seen less often in genital herpes in favor of HSV-1, due to recent changes in herpes infection epidemiology [31].

Recently, a nationwide cohort study compared the clinical features of 47 adults hospitalized for HSV-2 MM with 118 patients with single-episode HSV-2 meningitis. The clinical findings of headache, neck stiffness and photophobia were similar, but functional outcomes, evaluated by the Glasgow Outcome Scale, were more favorable in HSV-2 MM patients [12]. The same triad of symptoms are reported in our patient. Taking into consideration neurological sequelae, our patient had a favorable outcome, with symptom-free intervals between the episodes and without neurological permanent deficit, according to Bruyn’s criteria.

Although benign and spontaneous recovery is described in MM in the absence of antiviral therapy, our patient was treated with acyclovir, as commonly administered in aseptic meningitis. Regarding prophylaxis with oral antiviral therapy (0.5 g of valacyclovir twice daily), this was suggested for patients with recurrent HSV-2 meningitis, but no effect in terms of meningitis recurrence was observed [32].

## 4. Conclusions

This case underscores the importance of comprehensive diagnostic testing and a vigilant clinical approach to recognize this benign syndrome associated with a good prognosis. Further research is needed to evaluate the appropriate treatment or long-term prophylaxis for MM and the predisposing environmental or genetic factors in the development of the disease.

## Figures and Tables

**Table 1 microorganisms-12-01363-t001:** CSF findings during the patient’s hospital admissions.

	2004	2013	2016	2023	Normal Value
Protein (mg/dL)	92 (↑)	97 (↑)	79 (↑)	89(↑)	<50
Glucose (mg/dL)	45 (↓)	56 (=)	48 (↓)	57 (=)	50–80
WBC (/mm^3^)	70 (↑)	689 (↑)	106 (↑)	88 (↑)	0–5
Lymphocytes (%)	60% (=)	99% (↑)	99% (↑)	97.8% (↑)	60–70
HSV-2DNA (copies/mL)	/	<500	<500	1234	

CSF: cerebrospinal fluid; WBC: white blood cells; /: not performed; ↑: increased; ↓: reduced; =: normal.

**Table 2 microorganisms-12-01363-t002:** Case reports of Mollaret’s meningitis in immunocompetent patients (publication period: 2014–2023).

	No. Episodes	Age	Sex	CSF Parameters				Treatment	Clinical Manifestations
				WBC (/mm^3^)	Lymphocytes (%)	Glucose mg/dL	Protein (mg/dL)		
Edi et al., 2023 [13]	7	27	F	216	93	47	113	Vancomycin; Acyclovir	Malaise, headache, fever, neck pain
Grinney and Mohseni 2022 [14]	7	35	F	93	93	/	/	Ceftriaxone; Acyclovir; Gabapentin	Headache, nausea, vomiting, neck stiffness
Sehgal et al., 2021 [15]	2	83	M	103	92	58	113	Acyclovir	Fever, generalized weakness
Chand et al., 2021 [16]	6	88	M	139	82	46	122	Acyclovir	Headache, lethargy, altered sensorium
Park et al., 2021 [17]	6	35	F	649	94	40–70	<50	Acyclovir	Headache, neck pain, photophobia, binocular horizontal diplopia
Menon et al., 2020 [18]	4	50	F	6		162	102	Acyclovir; Phenytoin	Fever, agitated delirium, confused speech
	2	9	F	114	92	58	74	Acyclovir	Fever, headache, vomiting
Gadhiya and Nookala, 2020 [12]	4	44	F	/	/	102	32	Acyclovir	Headache, photophobia, nausea, neck pain
Kirkland et al., 2020 [19]	3	30	F	283		48	66	Acyclovir	Neck pain, stiffness, headache, malaise
Hait et al., 2020 [20]	6	50	F	66	>Monocytes	47	67	/	Nuchal rigidity, confusion, headache, minor cognitive changes
	12	78	F	987	>Monocytes	63	110	/	Nuchal rigidity, headache
Wright et al., 2019 [21]		44	F			35	231	Acyclovir; Vancomycin; Ceftriaxone	Headache, posterior neck pain, photophobia, nausea
Gluck et al., 2019 [22]	4	39	F	775	82	/	90		Headache, neck stiffness
Kawabori et al., 2019 [23]	4	34	M	212	91	71	48	Acyclovir; Ceftriaxone	Fever, headache, nausea and vomiting
Acosta et al., 2017 [24]	2	14	M	/	/	< 10	/	/	Headache, bilateral leg pain, lower limb weakness, diplopia
Govindarajan and Salgado 2016 [25]	2	70	M	175	75	50	70	Acyclovir	Aphasia, genital vesicles
Abou-Foul et al., 2014 [26]	/	40	M	/	/	/	/	Acyclovir	Headache, neck stiffness, photophobia, bilateral pedal paresthesia
Min and Baddley 2014 [27]	3	48	M	327	84	Normal	120	/	Fever, headache, neck stiffness, nausea and vomiting

CSF: cerebrospinal fluid; WBC: white blood cells; /: not reported; F: female; M: male. **Legend:** Patients with hematologic malignancies, immunodeficiencies, autoinflammatory genetic disorders and spinal tumors were not included. Meningitis caused by viruses other than HSV-2 were not included. Only reports in English were included from the literature. The age refers to the last episode/hospitalization of the patient. The medium recurrence of episodes was 4 years (range: 2–12), with a medium age of 45 (range: 9–88). A higher prevalence in women (63.6%) compared to men (36.4%), with a ratio of 11:7, was observed.

## Data Availability

The original contributions presented in the study are included in the article, further inquiries can be directed to the corresponding author.

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
