# Peer review of "Mollaret’s Meningitis due to Herpes Simplex Virus 2: A Case Report and Review of the Literature"

_microorganisms, 2024, doi:10.3390/microorganisms12071363_

Round 1

Reviewer 1 Report

Comments and Suggestions for Authors

Summary: Gabrielli and colleagues provide a case report that describes a patient with Mollaret’s meningitis associated with herpes simplex virus type 2 (HSV2) infection. Clinical findings are presented that describe 4 episodes of recurrent asceptic lymphocytic meningitis. HSV2 DNA was detected within CSF collected during 3 of 4 episodes using quantitative PCR assay. The authors provide a literature review of prior patients with Mollaret’s meningitis associated with HSV2 infection.

Review: This is a well written case report and review of the literature that provides an excellent historical background and clinical overview of Mollaret’s meningitis. The case report is significant because it nicely describes findings associated with 4 different recurrent episodes of the neurologic disease. That said, some concerns present themselves.

1. The authors correctly state that other viruses have been associated with Mollaret’s meningitis, most of them being herpesviruses. Specifically, the authors cite a 1991 paper by Yamamoto et al describing the detection of HSV type 1 (HSV1) DNA within CSF from a patient with the disease [Line 44]. The authors, however, should know that this was not the first time HSV1 was associated with Mollaret’s meningitis. Indeed, the first report came from Baringer’s group in 1982 when infectious HSV1 was recovered from CSF samples collected from a patient with Mollaret’s meningitis (4 recurrent episodes) (Steel et al, Ann Neurol 111: 17-21, 1982), and the authors should cite this first paper. In fact, this HSV1 isolate was found to have surprisingly unusual neurovirulence properties in mice when compared with other HSV1 neurologic isolates from brain tissues of patients with herpes simplex encephalitis (Dix et al, Infect Immun 40: 103-112, 1983).

2. Although the author’s state directly that their review is focused exclusively on HSV2-associated Mollaret meningitis, they are actually doing the reader (who might include clinical neurologists) a disservice by not including more information on HSV1-associated Mollaret’s meningitis. HSV1 and HSV2 are both neurotropic alpha herpesviruses that share 50% DNA homology, type-common glycoproteins immunologically, identical features of pathogenesis. and both are susceptible to ACV treatment. The authors provide a short paragraph on the possible association of recurrent episodes of Mollaret’s meningitis with recurrent HSV2 genital disease (or at least from reactivated virus from dorsal root ganglia) (Lines 173 – 176), but this should be expanded to provide more information on possible HSV1-associated Mollaret’s meningitis due to the significant similarities of HSV1 versus HSV2. More importantly, however, is the growing recognition that HSV1 is being associate with more cases of genital disease worldwide, so it is possible that clinicians will see a growing population of patients with HSV1-associated Mollaret’s meningitis.  

Author Response

Dear Collegue,

We would like to thank you for your helpful suggestions, you can find a point by point answer.

Regarding the first point:

The authors correctly state that other viruses have been associated with Mollaret’s meningitis, most of them being herpesviruses. Specifically, the authors cite a 1991 paper by Yamamoto et al describing the detection of HSV type 1 (HSV1) DNA within CSF from a patient with the disease [Line 44]. The authors, however, should know that this was not the first time HSV1 was associated with Mollaret’s meningitis. Indeed, the first report came from Baringer’s group in 1982 when infectious HSV1 was recovered from CSF samples collected from a patient with Mollaret’s meningitis (4 recurrent episodes) (Steel et al, Ann Neurol 111: 17-21, 1982), and the authors should cite this first paper. In fact, this HSV1 isolate was found to have surprisingly unusual neurovirulence properties in mice when compared with other HSV1 neurologic isolates from brain tissues of patients with herpes simplex encephalitis (Dix et al, Infect Immun 40: 103-112, 1983).

We added the two references you suggested in the introduction section. References number 3 and 4.

In addition we reported in the discussion section this sentence: “An association between HSV-1 and MM has been described in rare cases, the first report was by Steel et al [2], who identified HSV-1 in the CSF of a patient with four episodes of aseptic meningitis.”

We added also: “However, viral meningitis in immunocompetent adults is usually caused by HSV-2, while HSV-1 is more commonly associated with encephalitis [28]” because in the literature this association is always reported and can guide the clinician in the identification of the etiological agent even if both viruses are susceptible to acyclovir treatment.

 Regarding the second point:

Although the author’s state directly that their review is focused exclusively on HSV2-associated Mollaret meningitis, they are actually doing the reader (who might include clinical neurologists) a disservice by not including more information on HSV1-associated Mollaret’s meningitis. HSV1 and HSV2 are both neurotropic alpha herpesviruses that share 50% DNA homology, type-common glycoproteins immunologically, identical features of pathogenesis. and both are susceptible to ACV treatment. The authors provide a short paragraph on the possible association of recurrent episodes of Mollaret’s meningitis with recurrent HSV2 genital disease (or at least from reactivated virus from dorsal root ganglia) (Lines 173 – 176), but this should be expanded to provide more information on possible HSV1-associated Mollaret’s meningitis due to the significant similarities of HSV1 versus HSV2. More importantly, however, is the growing recognition that HSV1 is being associate with more cases of genital disease worldwide, so it is possible that clinicians will see a growing population of patients with HSV1-associated Mollaret’s meningitis.  

 We emphasized the correlation between HSV-1 and MM, as described in the first point. As requested we added that HSV-1 is being associated with more cases of genital herpes, adding this sentence:” However, HSV-2 is currently being seen less often in genital herpes in favor of HSV-1, due to recent changes in herpes infection epidemiology. [ Alareeki A, Osman AMM, Khandakji MN, Looker KJ, Harfouche M, Abu-Raddad LJ. Epidemiology of herpes simplex virus type 2 in Europe: systematic review, meta-analyses, and meta-regressions. Lancet Reg Health Eur. 2022 Dec 12;25:100558. doi: 10.1016/j.lanepe.2022.100558. PMID: 36818238; PMCID: PMC9929610.]”.

As you requested, we stressed  the literature data already available in the whole paper (introduction  and discussion).  However, it was not possible to add more details on possible HSV1-associated MM due to lack of literature data. The speculation of possible increase of HSV-1 MM is a good starting point but actually not supported by increased number of cases.

In addition:

  • at the end of the paper we added: Case Report Consent Form was signed by the patient.
  • The layout of Table 2 was fixed.

Reviewer 2 Report

Comments and Suggestions for Authors

The authors present an interesting case of a rare HSV-2 related but serious clinical condition, which indeed is a virological emergency. The case presentation is clear and scholarly, and the clinical managment as it should be. The importance of reactivation of HSV-2 is highlighted, which is highly relevant.

The literature overview is sound and embracing, and the overall manuscript is a timely reminder to perform proper diagnostics and also to think about rare events. Well done!

My only minor comment is on the formatting of the legend of table 2, there is no space between the legend and text. Please check.

Author Response

Dear Collegue,

We would like to thank you for your helpful suggestion, the legend of table 2 is fixed now, as you suggested

My only minor comment is on the formatting of the legend of table 2, there is no space between the legend and text. Please check.

In addition:

  • at the end of the paper we added: Case Report Consent Form was signed by the patient.
  • The layout of Table 2 was fixed.